# Ultra-Highly Efficient Removal of Methylene Blue Based on Graphene Oxide/TiO_2_/Bentonite Sponge

**DOI:** 10.3390/ma13040824

**Published:** 2020-02-11

**Authors:** Yuan Liu, Luyan Wang, Ni Xue, Pengxiang Wang, Meishan Pei, Wenjuan Guo

**Affiliations:** 1School of chemistry and chemical Engineering, University of Jinan, Jinan 250022, China; 20172120450@mail.ujn.edu.cn (Y.L.); 20172120409@mail.ujn.edu.cn (P.W.); chm_peims@ujn.edu.cn (M.P.); 2State Key Laboratory of Crystal Materials, Shandong University, Jinan 250100, China; 3Institute of Surface Analysis and Chemical Biology, University of Jinan, Jinan 250022, China; chm_guowj@ujn.edu.cn

**Keywords:** graphene oxide, sponge, bentonite, titanium dioxide, photocatalytic

## Abstract

An ultra-highly efficient Graphene Oxide/TiO_2_/Bentonite (GO/TiO_2_/Bent) sponge was synthesized using an in situ hydrothermal method. GO/TiO_2_/Bent sponge with a GO mass concentration of 10% exhibited the highest treatment efficiency of methylene blue (MB), combining adsorption and photocatalytic degradation, and achieved a maximum removal efficiency of 100% within about 70 min. To further prove the ultra-high removal capacity of the sponge, the concentration of MB in water increased to ten times the original concentration. At so high a MB concentration, the removal rate was still as high as 80% in 90 min. The photocatalytic mechanism of GO/TiO_2_/Bent sponge was discussed through XPS, PL and radicals quenching experiments. Here Bent can immobilize TiO_2_ and react with a photo-generated hole to increase the amount of hydroxyl radical; effectively enhancing the degradation of MB.GO sponge enlarges the sensitivity range of TiO_2_ to visible light by increasing the charge separation of TiO_2_ and reducing the recombination of photo-generated electron–hole pairs. Additionally, GO sponge with an interconnected porous structure provides an effective platform to immobilize TiO_2_/bent and makes them be easily recovered. The as-prepared sponge develops a simple and cost-effective strategy to realize the ultra-highly efficient treatment of dyes in wastewater.

## 1. Introduction

Dyes are commonly used in many industrial applications, including textiles, printing, plastics, leather and papermaking. When these pollutants are discharged into the environment, they present a serious challenge to aquatic life, the food chain and human health [1,2,3,4]. Owing to the difficulty of dye degradation, it is difficult for normal treatment by biochemical methods to meet the requirements for their treatment. Photocatalytic decomposition of organic compounds has been widely studied to elevate the degradation efficiency of dyes in aqueous solution, wherein TiO_2_ has received much attention and has been reported in various studies [5,6,7,8]. TiO_2_ is one of the most popular types of photocatalyst, and it is utilized in many fields, such as wastewater degradation and energy fuel, because of its high efficiency, nontoxicity, good chemical stability, and low cost [9,10].

However, because of its 3.2 eV electronic band gap, TiO_2_ is only sensitive to ultraviolet light, and cannot take full advantage of solar energy. In addition, TiO_2_ usually also has a small surface area and low absorbability, and its photocatalytic effect is low in solutions [11,12,13,14,15]. Additionally, commercial nano-TiO_2_ is of super-hydrophilicity, and easily agglomerates in water. The conglomeration of TiO_2_ particles not only weakens its photocatalytic performance, but is also unfavorable to its reutilization. In recent years, many efforts have been directed towards facilitating the use of this cost-effective photocatalyst [16,17,18]. For example, graphene oxide (GO) or reduced graphene oxide (RGO) has been used to modify TiO_2_ for photocatalytic degradation of organics, such as MB and Rhodamine B (RhB), by improving charge separation of TiO_2_ and reducing the combination of photo-generated electron–hole pairs [19,20,21,22,23]. However, the photocatalytic effect is far from satisfactory. Wang et al. prepared several graphene-titanium composites by a sol–gel method using titanium isopropoxide as Ti-precursors [24]. In the first 30 min (dark reaction), the maximum removal rate of MB reached only about 31%. Meanwhile, MB was completely degraded in about 90 min (Content of MB: 10 mg/L, 200mL; Mass of photocatalyst: 100 mg). Li et al. obtained TiO_2_-graphene nano-composite photocatalyst using a facile one-step hydrothermal method [25]. Under simulated sunlight, the maximum degradation rate is only 63% in 65 min (content of MB: 10 mg/L, 40 mL; mass of photocatalyst: 30 mg). Fan’s group reported that TiO_2_ doped by carbon could extend TiO_2_ light absorption cut off wavelength. However, the agglomeration of TiO_2_ nanoparticles on graphene prohibited the direct chemical contact between the two components [26]. In addition, Su-Il researcher groups used a TiO_2_ nanotube array photoanode coupled with a conventional bioanode, achieving simultaneous degradation of methylene blue (MB) dye and improved power generation [9].

In recent years, clays, such as bentonite, rectorite and kaolinite, have been widely used in adsorption fields [27,28,29]. These natural and low-cost materials provide layered structures, large surface areas and a high cation exchange capacity [30,31,32,33]. It was reported in our previous paper that the clay bentonite can exhibit an improved adsorption capacity and reusability after modification [34]. Additionally, in the early work of our group [35], three-dimensional GO sponge was found to have a much larger surface area than two-dimensional GO sheets. Meanwhile, other groups have also performed the similar works on three-dimensional GO sponge [36,37].

In this work, to improve the adsorption and photocatalytic properties of TiO_2_, as well as the reusability of the material for removing MB in water, the sponge-like composite made of GO, TiO_2_ and bentonite is designed (see Figure 1). Firstly, in order to further enhance the degradation ability of photocatalyst to organic pollutants by a simple and cost-effective method, bentonite is introduced to load TiO_2_. The important role of bentonite and the photocatalytic mechanism are expounded in detail. On the one hand, bentonite, as an excellent carrier with a large surface area, can effectively immobilize TiO_2_ nanoparticles and prevent TiO_2_ from agglomeration. On the other hand, the surface of bentonite has abundant hydroxyl functional groups. TiO_2_, as a kind of semiconductor, will produce electron–hole pairs under the excitation of photons. The photo-generated holes have strong oxidation properties and can oxidize the hydroxyl groups on the bentonite surface to hydroxyl radicals and thus improve the photocatalytic efficiency. In addition, bentonite can also strongly adsorb pollutants on its surface, particularly organic cationic dyes, playing a role as a medium, which is also advantageous to the photocatalytic process.

Secondly, as mentioned above, the introduction of GO sponge, which has an interconnected structure to promote the transfer of electrons, favors the increase of TiO_2_ photocatalytic properties by improving the charge separation of TiO_2_, reducing their combination of photo-generated electron–hole pairs and strengthening the response of TiO_2_ to the sun light.

The third, three-dimensional GO sponge provides plenty of space to accommodate as many TiO_2_/bent particles as possible. In addition, it acts as an efficient platform for immobilizing the particles and prevents them from aggregation and losing in water. In addition, more notably, as a whole sponge which can be easily recovered, the GO/TiO_2_/bent composite exhibits an excellent reusability which is a difficult problem that troubles us for a long time in this field.

## 2. Experimental

### 2.1. Materials

Graphite flakes (>99.7%, weight percent) were purchased from Qingdao Chemical Reagent Co. Ltd. (Qingdao, China). Sodium nitrate, sulphuric acid (98%), potassium permanganate, hydrochloric acid (36%), barium sulfate, and TiO_2_ (P25) were purchased from Shanghai Maclin Biochemistry Technology Co., Ltd. (Shanghai, China). All chemicals were used as received without further purification.

### 2.2. Preparation of GO Sponge

GO was prepared based on Hummers’ method [35,38]. Graphite flakes (0.6 g) and sodium nitrate (1 g) were added slowly into concentrated sulfuric acid (35 mL), maintaining the temperature of the ice water bath between 0 and 4 °C. Then, the potassium permanganate was added to the solution very slowly, while simultaneously maintaining the temperature at no more than 20 °C. Next, the solution was put into a water bath of 35 °C, with stirring, for 2 h. After that, 150 mL deionized water was slowly added to the solution and the temperature was maintained at 98 °C for 15 min. The liquid was further diluted with 200 mL water, and stirred for 2 h. At the same time, 10 mL of hydrogen peroxide was added, until the color of the sample turned golden. Subsequently, the solution was stirred for 3 h and the supernatant was poured out. Hydrochloric acid solution (V_HCl_: V_water_ = 1:10) and water were used to wash the golden solutions several times until the pH was greater than 5. Then, the golden sample was diluted to 4 mg/mL with water and the 4 mg/mL golden sample was diluted with 50 mL water. The suspension was placed into a 100 mL Teflon-sealed autoclave and kept at 180 °C for 10 h. Finally, the composite was freeze-dried. Then, the three-dimensional GO sponge was successfully prepared.

### 2.3. Preparation of GO/TiO_2_ and TiO_2_/Bent

The GO/TiO_2_ sample was prepared as followed. The bright-yellow GO solution was diluted to 4 mg/mL by water and then 50 mL of such sample was used to disperse TiO_2_. In GO/TiO_2_ samples, GO loading was kept at 5%, 10%, 15% by adjusting TiO_2_ amount to 1.33, 2, and 4 g, respectively. The suspension was placed in a 100 mL Teflon-sealed autoclave and kept at 175 °C for 10 h. Finally, the composite was obtained by freeze-drying.

TiO_2_/Bent sample was prepared as follows. Bent (0.5 g) and TiO_2_ (0.5 g) were mixed into water (20 mL), kept in a water bath at 60 °C, and stirred for 2 h. Then, the dispersion was aged for 12 h at 70 °C. The TiO_2_/Bent sample was obtained by centrifuging, washing and drying at 80 °C for 12 h.

### 2.4. Preparation of GO/TiO_2_/Bent Sponge

The bright-yellow GO solution was diluted to 4 mg/mL with water and then 50 mL of this sample was used to disperse TiO_2_/Bent. In GO/TiO_2_/Bent samples, GO loading was kept at 5%, 10%, and 15% by adjusting TiO_2_/Bent amount to 1.33, 2, and 4 g, respectively. The suspension was placed in a 100 mL Teflon-sealed autoclave and kept at 175 °C for 10 h. Finally, the composite was obtained by freeze-drying, and then fully ground for further characterization.

### 2.5. Photocatalytic Tests

All the adsorption experiments were conducted three times and the averaged values were used as the experimental data to make the results reliable. Photocatalytic performances of various catalysts were evaluated by the photo-degradation of methylene blue (MB) under artificial solar light (λ = 220–1050 nm). In a typical process, aqueous solution of MB (10 mg/L or 100 mg/L, 200 mL) and the photocatalysts (10 mg) were put into a beaker under constant stirring. The intensity of artificial solar light was 86 mW/cm^2^ in each test. Firstly, the reaction solution was stirred for 30 min in dark to achieve adsorption equilibrium. The photocatalytic reaction was started by turning on a Xeon lamp. The suspension solution was centrifuged, and the MB solution was analyzed at 10-minute intervals to obtain the diffuse reflectance spectra by a UV-vis (Beijing Persee General Instrument Co., Ltd., Beijing, China) spectrophotometer at a wavelength of 664 nm.

### 2.6. Characterization of Materials

The crystalline structure of samples was analyzed by Bruker AXS D8 Focus X-ray diffraction (XRD, Bruker AXS, Karlsruhe, Germany) operating at 40 kV and 40 mA. The scan angle 2θ varied from 5° to 80°, and the scan speed was 0.04°·s^−1^. FTIR analysis was performed on a Bruker VECTOR-22 and the spectra were collected over the spectral range of 500–4000 cm^−1^. Scanning electron microscopy (SEM, JSM-6700F, JEOL Ltd., Sartorius, Göttingen, Germany) was used to evaluate the morphology and size information of the samples. X-ray photoelectron spectroscopy (XPS, ESCALAB 250, Thermo Fisher Scientific, Waltham, MA, USA) can be used in qualitative and quantitative analysis to characterize the surface composition of samples. Photoluminescence (PL) spectra were performed by F-4600 FL (Hitachi, Tokyo, Japan) using excitation wavelength of 320 nm. Nitrogen adsorption–desorption isotherms were measured at −196·°C with a Tristar II 3020 (Micromeritics Inc., Norcross, GA, USA).

## 3. Results and Discussion

### 3.1. Characterization of Materials

The FTIR spectra of Bent, GO, TiO_2_, 10% GO/Bent and 10% GO/TiO_2_/Bent are shown in Figure 2. The band at 3432 cm^−1^ is ascribed to a stretching vibration of the hydroxyl group (O–H) of the adsorbed water molecules in all the samples. For Bent, the band at 1638 cm^−1^ is due to the bending vibrations of the H–O–H bonds of the water molecules intercalated in the clay mineral. The strong absorption peak at 1033 cm^−1^ is the stretching vibration absorption peak of the Si–O–Si group, while the absorption peaks at 791, 624 and 521 cm^−1^ are caused by the deformation and bending vibration of the Si–O bond [39].

For GO, C=O stretching at 1725 cm^−1^, O–H bending and aromatic C=C stretching at 1621 cm^−1^, tertiary C–OH stretching at 1390 cm^−1^ and epoxy C–O stretching at 1068cm^−1^ are clearly observed, suggesting the presence of carboxyl, hydroxyl and oxygenation functional groups. For TiO_2_, the strong absorption bands observed at low frequency 684 cm^−1^ corresponds to the vibration of Ti–O–Ti bonds [40]. GO/Bent exhibits the characteristic peaks of skeleton vibration of GO, the deformation and bending vibration of Si–O bond at 791, 624 and 521 cm^−1^, respectively. For 10% GO/TiO_2_/Bent, the characteristic peaks of Si–O bond of Bent and Ti–O–Ti bond of TiO_2_as well as the characteristic peak of functional group of GO are found.

The characteristic skeletal vibration peaks of GO sponge marked by rectangle are also observed in FTIR spectra [25]. Due to the strong combination between TiO_2_ and Bent, the main skeletal vibration peaks of GO shift from 2928 cm^−1^ to 2924 cm^−1^ (See Appendix A). It is indicated that the chemical interaction between GO and TiO_2_/Bent.

Diffuse reflectance spectra of TiO_2_, Bent, TiO_2_/Bent, 10% GO/TiO_2_ and 10% GO/TiO_2_/Bent are shown in Figure 3. As can be seen, Bent is almost transparent in the wavelength range longer than 350 nm [16]. TiO_2_/Bent displays an obvious red shift of about 19 nm in the absorption edge compared to TiO_2_. This may be attributed to the effects of Bent or some elements of Bent which may be doped into TiO_2_ [17]. Whereas the photocatalysts of TiO_2_, 10% GO/TiO_2_ and 10% GO/TiO_2_/Bent show the absorption edge at about 394, 441 and 467 nm, respectively. Furthermore, the figures indicate that the threshold of photocatalysts (10% GO/TiO_2_ and 10% GO/TiO_2_/Bent) display obvious red shifts to longer wavelength regions when compared with pure TiO_2_. The experimental results show that GO has a significant influence on the band gap of TiO_2_. As a result, solar light is utilized more efficiently by 10% GO/TiO_2_/Bent than other compared photocatalysts. This may be attributed to the formation of Ti–O–C chemical bonding in the composites [25,41,42].

Figure 4 shows the XRD patterns of Bent, GO, TiO_2_, TiO_2_/Bent, 10% GO/Bent and 10% GO/TiO_2_/Bent. The peaks at 2θ = 11.16° and 42.51° are assigned to the (002) and (100) reflections of GO. Clearly, all the peaks for TiO_2_ are readily indexed to the anatase phase of TiO_2_ (JCPDS No. 21-1272). Furthermore, the absence of the typical diffraction of GO stacking layers in the composites (10% GO/Bent and 10% GO/TiO_2_/Bent) might be attributed to the disruption and well exfoliation of GO in the composite or its low diffraction intensity [41,43]. The acute diffraction peak at 2θ = 6.68° in Bent is the (001) reflection, which indicates the existence of a layered structure. However, the TiO_2_/Bent sample does not have a sufficiently ordered and oriented silicate layer structure to show the (001) peak, as does the sample of GO/TiO_2_ [44]. The typical characteristic peak of GO and Bent do not appear in the composite. This is because the diffraction peak of TiO_2_ is so strong that the diffraction peaks of GO and Bent are submerged [45].

Figure 5 displays SEM images of GO, 10% GO/TiO_2_, 10% GO/Bent, and 10% GO/TiO_2_/Bent. The inset of each picture corresponds to the macroscopic columnar sponge. As can be seen from Figure 5a, GO appears as honeycomb-like sponge structure with large holes. After TiO_2_ is added, the surface of holes in GO sponge is covered by TiO_2_.The structure of 10% GO/TiO_2_/Bent is similar to that of 10% GO/Bent. The sponge-like structure of GO is beneficial for the transmitting of photo-generated electrons between GO layers and provides an ideal support for the deposition of TiO_2_ and Bent particles [46].

### 3.2. Photocatalytic Degradation

The removal rate of MB by GO/TiO_2_/Bent composite with different GO proportions was evaluated and the results were shown in Appendix A. It can be seen that the removal efficiency by 10% GO/TiO_2_/Bent is the best. Therefore, 10% GO/TiO_2_/Bent sponge is selected for the next experiments. In parallel experiments, the MB removal rates by TiO_2_, GO, TiO_2_/Bent, 10% GO/TiO_2_, 10% GO/TiO_2_/Bent, respectively, are obtained under simulated sunlight illumination after 30 min dark adsorption, and the results are shown in Figure 6. It can be seen that 10% GO/TiO_2_/Bent composite shows significant progress in the adsorption and degradation of MB. In the first 30 min (dark reaction), the removal rate of MB is about 99.7%, and subsequently increases to up to 100% at 70 min under sunlight. Additionally, in the first 30 min, TiO_2_/Bent exhibits a better adsorption capacity of MB than that of 10% GO/TiO_2_. However, with the increase of time, 10% GO/TiO_2_ shows good degradation ability. BET values of each catalyst are not related to the photocatalytic dye degradation activity (Appendix A and Appendix A).

As mentioned above, in the first 30 min, the adsorption of MB by 10% GO/TiO_2_/Bent is particularly prominent, with a removal rate as high as 99.7%. Therefore, the photocatalytic ability of the sponge to MB under light is not well demonstrated. Here, to better present the effectiveness of photocatalysis, the concentration of MB is specifically increased to 100 mg/L (Figure 7), which is ten times the original MB concentration shown in Figure 6. It can be seen from Figure 7 that the adsorbent of 10% GO/TiO_2_/Bent sponge presents not only a high adsorption effect on MB, but also a highly photocatalytic efficiency under light. In general, 10% GO/TiO_2_/Bent sponge exhibits an excellent removal rate of MB, which is still as high as 80% in 90 min at so high a concentration.

Figure 8 shows the evolutions of absorption spectra of MB solution in the presence of 10% GO/TiO_2_/Bent sponge under sunlight at different exposure time. It can be clearly seen that the characteristic absorption peak of MB solution at 664 nm is significantly decreased in intensity with the increasing irradiation time. In addition, the absorption peak disappears when the exposure time is increased to 70 min, along with the color fading of the solution, indicating that MB is completely removed, which is consistent with the result shown in Figure 6.

Table 1 shows the removal rates of MB by different materials previously used. It can be observed that the removal rate of MB by 10% GO/TiO_2_/Bent is much higher than that of other materials listed, indicating that 10% GO/TiO_2_/Bent has great potential application in MB removal from water.

### 3.3. Photocatalytic Mechanism

Based on the experimental data listed above, it is known that 10% GO/TiO_2_/Bent exhibits excellent removal efficiency of MB based on both adsorption and photocatalytic process. To explore the mechanism of photocatalysis and detect the surface elements of holes in sponge structure, XPS analysis is carried out on the GO/TiO_2_/Bent sample. Figure 9 is the full spectrum of XPS before and after photocatalytic reaction of the sample. After adsorption and photocatalysis, the surface elements of the composite change slightly. There is a small decrease of the content of Si (from 16.64% to 14.93%) and Al (from 5.2% to 4.6%) on the surface of the composite after reaction, due to the slight dissolution of Bent on the outer surface during the reaction. In fact, the reaction, which occurs on Bent surface by consuming the hydroxyl groups to produce •OH, will accelerate the dissolution of Bent on the outer surface. Figure 10 is the XPS spectrum of O element in 10% GO/TiO_2_/Bent. Binding energy 532.2 eV is the characteristic peak of –OH on the catalyst surface [62,63]. The binding energy of 529.8 eV is Ti–O–Si bond, indicating that Ti–O–Si bond is formed between TiO_2_ and Bent. It is the binding of chemical bond and strong loading of TiO_2_ on the surface of Bent, which is favorable for charge transfer [64,65].

It is known that photoluminescence (PL) spectra are often employed to reveal the separation performances of electron–hole pairs in semiconductors [66]. Therefore, the separation properties of photo-generated electron–hole pairs of composites were further studied by PL. Figure 11 shows the PL emission spectra of TiO_2_ and GO/TiO_2_/Bent composites. It can be found that the spectra of GO/TiO_2_/Bent samples appear to be similar with that of TiO_2_, which means that GO and Bent have not induced new photoluminescence. In addition, the PL intensities of the composites are weakened in comparison to pure TiO_2_ and gradually decrease with the increasing of GO content. When GO content is less than 10%, the fluorescence intensity of GO/TiO_2_/Bent decreases with the increasing of GO content, which indicates that GO can rapidly accept electrons generated by photo excitation of TiO_2_, and effectively inhibit the recombination of electron–hole pairs. However, the PL intensity of 15% GO/TiO_2_/Bent is lower than that of TiO_2_, but higher than those of 5% and 10% GO/TiO_2_/Bent. When GO content is more than 10%, the fluorescence intensity of GO/TiO_2_/Bent increases with the increasing of GO content, indicating that excess GO has no positive effect on the light effect of composite. Moreover, excessive GO will cause the mask effect, which is not favorable for enhancing the photocatalytic activity of photocatalysts toward MB. To sum up, the photoluminescence (PL) spectra verify that GO/TiO_2_/Bent sponges exhibit excellent electron transport ability.

The whole mechanism of photocatalytic degradation is shown in Figure 12 and can be described by the following equations.
(1)$$\mathrm{GO}/\mathrm{TiO}2/\mathrm{Bent}+\mathrm{hv}\overset{}{\to }{\mathrm{GO}/\mathrm{TiO}2/\mathrm{Bent}(e}^{-}+h^{+})$$
(2)$$e^{-}+O_{2}\overset{}{\to }•O_{2}{}^{-}$$
(3)$$h^{{}_{+}}+\mathrm{Bent}\equiv M-\mathrm{OH}\overset{}{\to }•\mathrm{OH}$$
(4)$$h^{{}_{+}}+H_{2}O\overset{}{\to }•\mathrm{OH}$$
(5)$$O_{2}+{2H}^{+}+{2e}^{-}\overset{}{\to }H_{2}O_{2}$$
(6)$$h^{{}_{+}}+\mathrm{MB}\overset{}{\to }\mathrm{Oxidation}\;\mathrm{products}$$
(7)$$e^{-}+\mathrm{MB}\overset{}{\to }\mathrm{Reduction}\;\mathrm{products}$$
(8)$$•\mathrm{OH}+\mathrm{MB}\overset{}{\to }{\mathrm{CO}}_{2}+H_{2}O$$
(9)$$•O_{2}{}^{-}+\mathrm{MB}\overset{}{\to }{\mathrm{CO}}_{2}+H_{2}O$$

The organic pollutant MB molecule is first adsorbed to the surface of GO/TiO_2_/Bent and then decomposed in situ from the surface of the composite. The three-dimensional GO sponge has excellent electronic transmission ability, which can effectively prevent the electron–hole pair recombination of TiO_2_ and improve the photocatalytic efficiency. Bentonite not only presents strong adsorption ability, but also provides a large number of hydroxyl groups on its surface, which can react with the holes (Equation (3)) to increase the production of •OH. These •OH can effectively attack the organic dyes adsorbed on the composite surface. Hydroxyl radical has strong oxidation ability and can degrade organic pollutants into CO_2_ and H_2_O, which plays an important role in photocatalytic reaction [67,68]. Therefore, Equation (3) based on Bent is especially crucial in the mechanism.

To verify whether 10% GO/TiO_2_/Bent produces •OH in photocatalytic reaction to improve photocatalytic efficiency, tert-butanol and isopropanol, which can selectively quench hydroxyl radical, were added to MB solution in this experiment to analyze the photolysis behavior under xenon lamp irradiation. To study the photo-degradation reaction after reaching adsorption equilibrium, the concentration of MB was selected as 100 mg/L, and the control experiment was carried out under the same conditions. The first group of MB solution did not contain any reagents and was used as blank control group; the second group of MB solution was combined with tert-butanol (40 mmol/L); the third group of MB solution was combined with isopropanol (40 mmol/L). The experimental results are shown in Figure 13. It can be seen that the degradation rate of MB is reduced by adding tert-butanol and isopropanol, proving that •OH produced by 10% GO/TiO_2_/Bent participates in the degradation of MB under light irradiation.

## 4. Conclusions

In this work, the novel GO/TiO_2_/Bent sponge was synthesized successfully by a facile hydrothermal method. It was found that the removal rate of 10% GO/TiO_2_/Bent is better than many other materials reported, and MB can be completely removed within 70 min. It is remarkable that even at a high MB concentration of ten times the original one, the sponge still exhibits an excellent removal rate of 80% within 90 min. Both Bent and GO sponge play an important role in the adsorption and degradation of the dyes. The surface of Bent is rich in hydroxyl groups, promoting the formation of •OH and enhancing the ability to degrade organic pollutants through photo-generated hole. Also Bent can immobilize nano-TiO_2_ and prevent its agglomeration. Three-dimensional GO sponge exhibits excellent electron transport ability, which can effectively prevent the combination of TiO_2_ electron–hole pairs and improve the photocatalytic efficiency. Significantly, its sponge-like structure not only provides a large specific surface area and more adsorption sites for TiO_2_/Bent, but also benefits the recycle of the composite. Because as a whole sponge it can be taken out of the solution, avoid the loss of TiO_2_ and Bent and then be easily reused. The results show that the removal rate is better than that of single-component and bi-component systems, which indicates that the enhanced removal performance presented by 10% GO/TiO_2_/Bent is not a simple superposition of the performance of each component, but a result of synergistic action among three components. In summary, the synthesized GO/TiO_2_/Bent sponge has great application prospects for the removal of pollutants from water.

## Figures and Tables

**Figure 1 materials-13-00824-f001:**
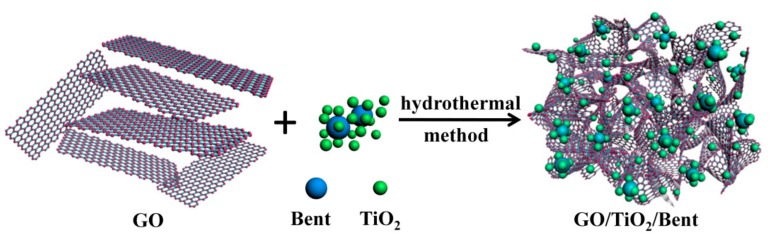
Scheme for the preparation of GO/TiO_2_/Bent sponge.

**Figure 2 materials-13-00824-f002:**
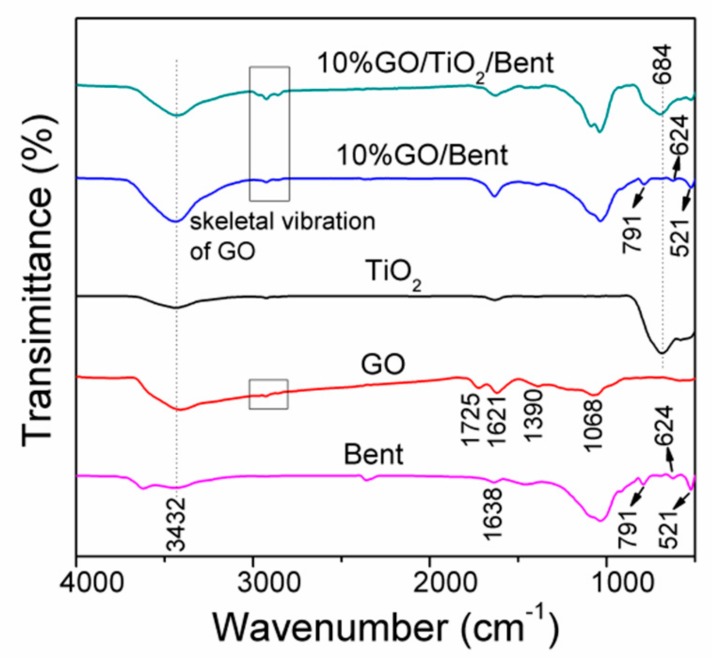
FTIR spectra of Bent, GO, TiO_2_, 10% GO/Bent and 10% GO/TiO_2_/Bent sponges.

**Figure 3 materials-13-00824-f003:**
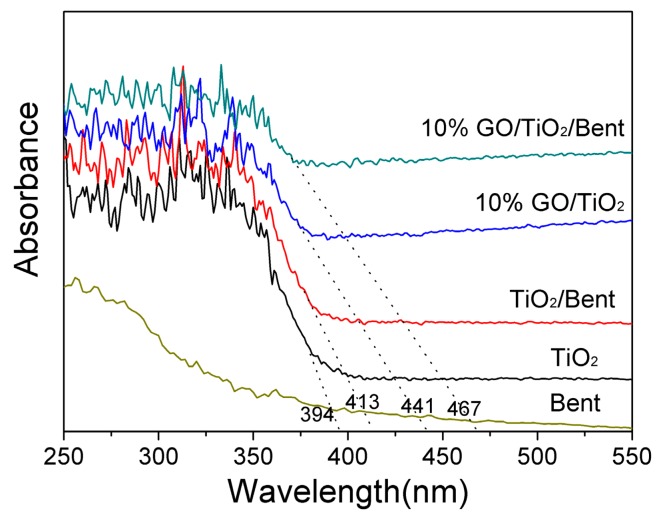
Diffuse reflectance absorption spectra of Bent, TiO_2_, TiO_2_/Bent, 10% GO/TiO_2_ and 10% GO/TiO_2_/Bent sponge.

**Figure 4 materials-13-00824-f004:**
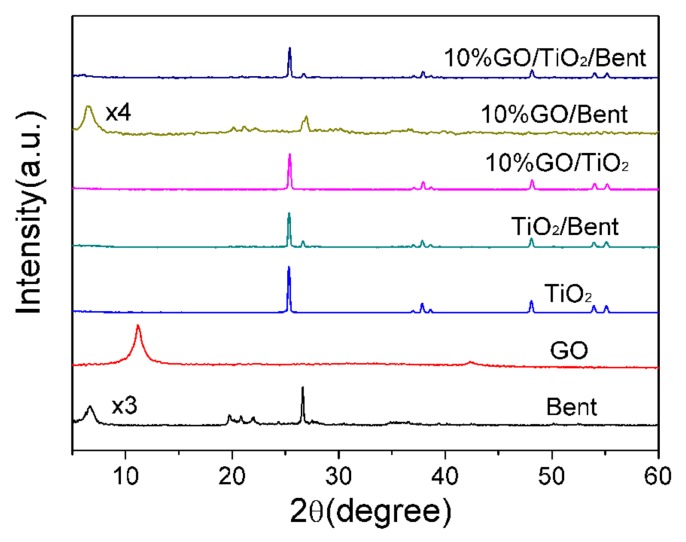
XRD patterns of Bent, GO, TiO_2_, TiO_2_/Bent, 10% GO/TiO_2_, 10% GO/Bent and 10% GO/TiO_2_/Bent.

**Figure 5 materials-13-00824-f005:**
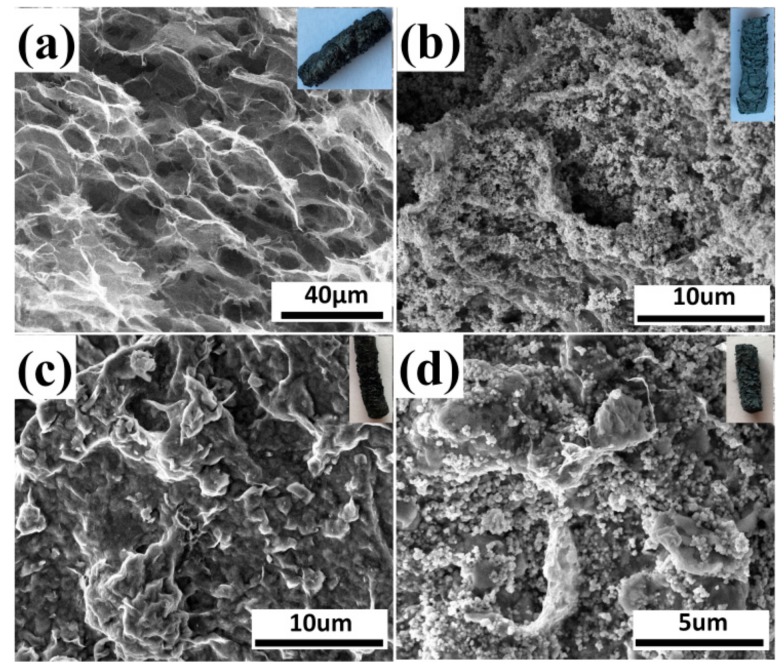
SEM images of (**a**) GO, (**b**) 10% GO/TiO_2_, (**c**) 10% GO/Bent, and (**d**) 10% GO/TiO_2_/Bent. Insets are pictures of macroscopic columnar sponge.

**Figure 6 materials-13-00824-f006:**
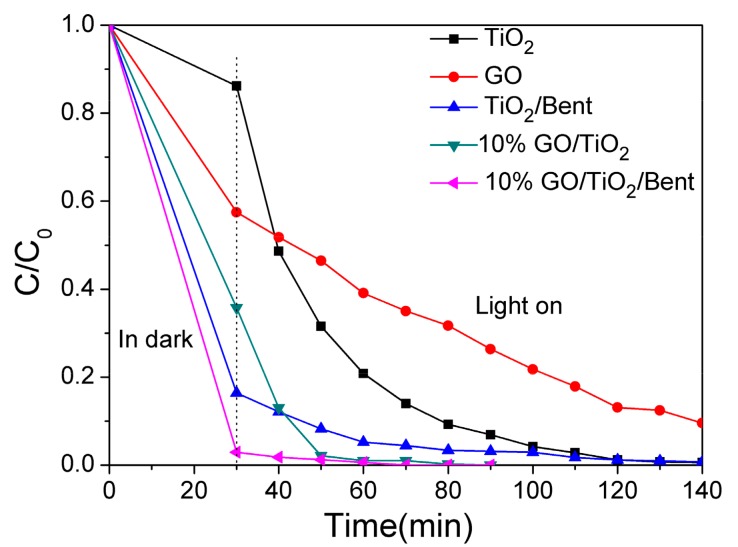
Removal rates of MB by different materials of TiO_2_, GO, TiO_2_/Bent, 10% GO/TiO_2_ and 10% GO/TiO_2_/Bent sponge, respectively (Concentration of MB: 10 mg/L in 100mL MB solution; Mass of materials: 50 mg).

**Figure 7 materials-13-00824-f007:**
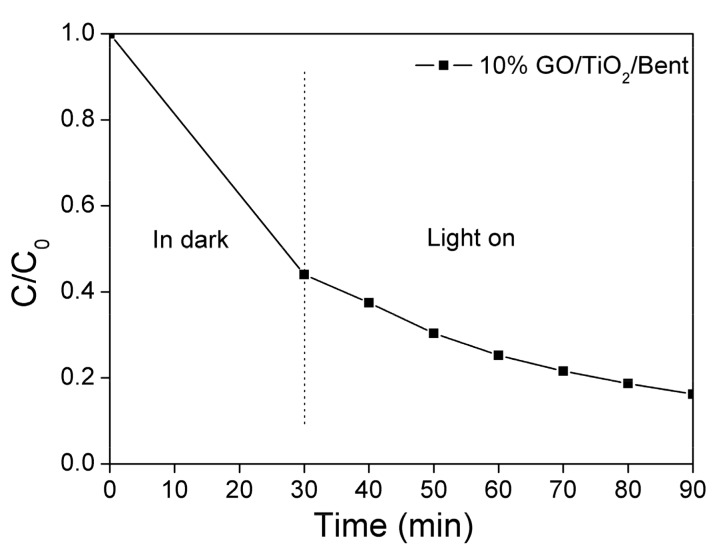
Removal rate of MB in solution with a higher concentration by 10% GO/TiO_2_/Bent (Concentration of MB: 100 mg/L in100 mL MB solution; Mass of 10% GO/TiO_2_/Bent: 50 mg).

**Figure 8 materials-13-00824-f008:**
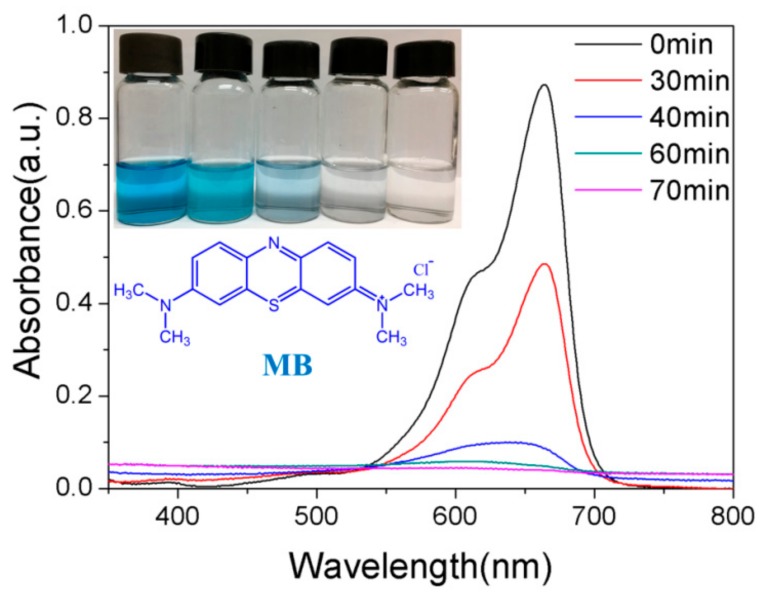
The evolutions of absorption spectra of MB in the presence of 10% GO/TiO_2_/Bent sponge under sun light at different time (Content of MB:10 mg/L, 200 mL; Mass of 10% GO/TiO_2_/Bent: 100 mg).

**Figure 9 materials-13-00824-f009:**
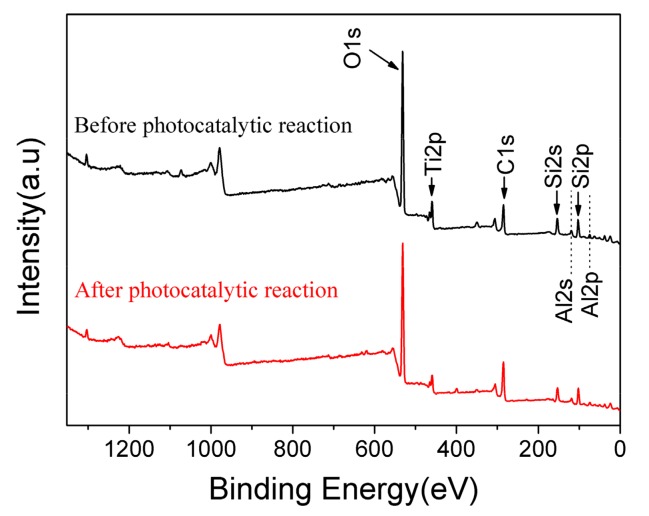
XPS survey spectrum of 10% GO/TiO_2_/Bent sponge before and after photocatalytic reaction.

**Figure 10 materials-13-00824-f010:**
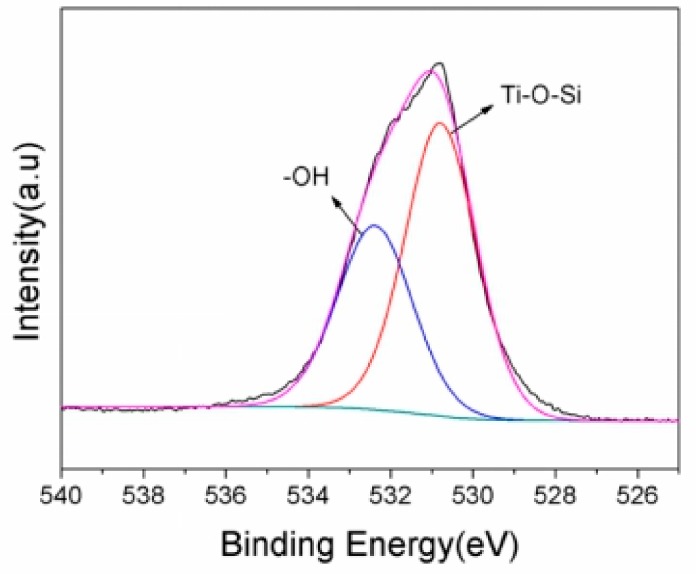
XPS spectra of O element in 10% GO/TiO_2_/Bent sponge.

**Figure 11 materials-13-00824-f011:**
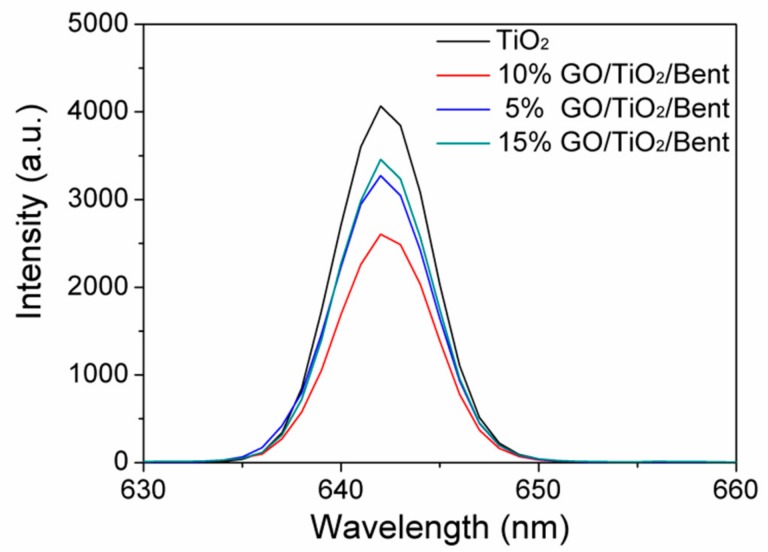
Photoluminescence (PL) spectra of TiO_2_ and GO/TiO_2_/Bent sponges.

**Figure 12 materials-13-00824-f012:**
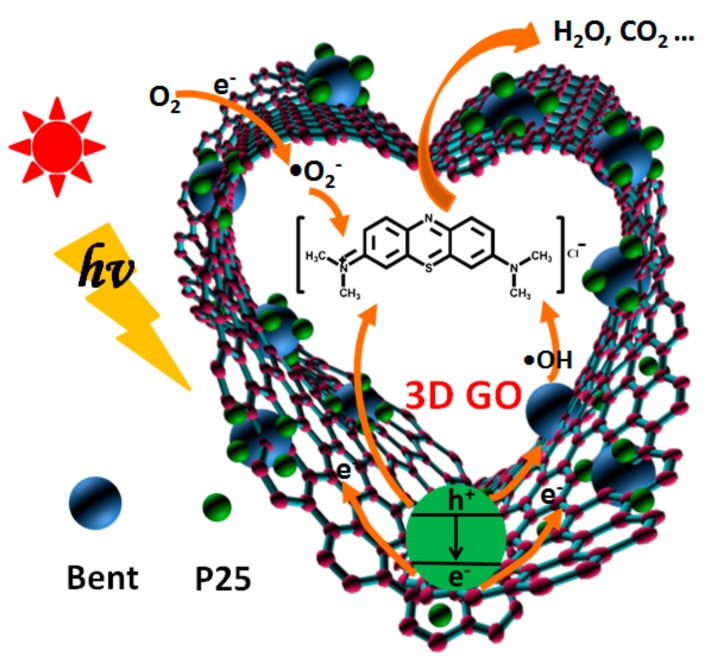
The photocatalytic degradation mechanism of MB based on GO/TiO_2_/Bent sponge.

**Figure 13 materials-13-00824-f013:**
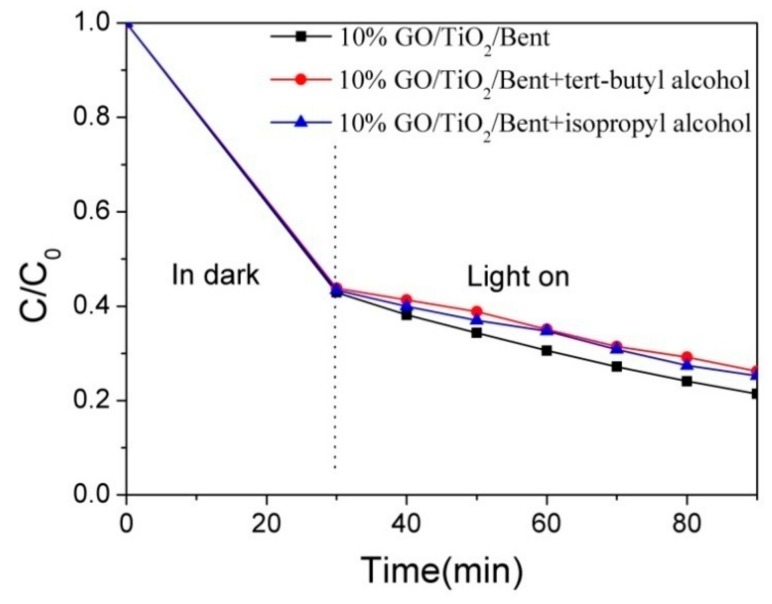
Effect of radical quenching agents on MB degradation by 10% GO/TiO_2_/Bent (Concentration of MB: 100 mg/L).

**Table 1 materials-13-00824-t001:** The removal rate of MB by various materials.

Number	Sample	Removal Rate (%)	Removal Time (min)	Reference
1	wsGNS	96a 99.9b	90a 120b	[47]
2	Hemin-functionalized graphenehydrogel (Hem/GH)	96	180	[48]
3	TiO_2_-Nb/N	93	120	[49]
4	SiO_2_-WO_3_	100	120	[50]
5	WO_3_/grapheme nanocomposite (WO_3_-G)	99	480	[51]
6	graphene-V_2_O_5_nanocomposite (GO-V_2_O_5_)	99	90	[52]
7	P25-GO	>99	240	[53]
8	TiO_2_-GO	>80	300	[54]
9	reduced graphene/manganese oxide (rGO/MnO_2_ hybrid)	66	5	[45]
10	zinc porphyrin functionalized graphene quantum dots (GQDs/ZnPor)	95	60	[55]
11	Mn_3_O_4_ decorated graphene oxide	99	200	[56]
12	TiO_2_-graphene	90	150	[57]
13	Ag-S/PEG/TiO_2_	77.51	120	[58]
14	water-soluble grapheme nanosheets (wsGNS)	98	75	[59]
15	electrospun porous fibers EPF(2/1)-TiO_2_	100	120	[60]
16	granular activated carbon doped with iron (Fe-GAC)	94	200	[61]
17	hybrid-MFC	82.79	210	[62]
18	GO/TiO_2_/Bent	100	70	present study

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
