# Peer review of "Ultra-Highly Efficient Removal of Methylene Blue Based on Graphene Oxide/TiO2/Bentonite Sponge"

_materials, 2020, doi:10.3390/ma13040824_

Round 1
Reviewer 1 Report
In the manuscript, the authors describe the preparation of graphene oxide/TiO2/bentonite sponge for the highly efficient removal of an organic pollutant by light irradiation. These results will be helpful and informative for the researchers in the field of materials chemistry.
Whereas the reviewer thinks that the authors’ study in this manuscript is quite interesting, suggestive, and well-organized, some descriptions are not enough. The authors’ manuscript is not suitable for publication in “Materials” in the present form.
From these considerations, the reviewer recommends accepting for publication in " Materials," if the following issues are resolved.
Figure S1: Why did 10% GO/TiO2/Bent sponge show better absorption of MB comparing to other GO/TiO2/Bent sponges in the dark? Figure 6: In the dark, why did the removal of MB by GO/TiO2/Bent sponge become better than other materials? In Figure 7, why did the slope of the line decrease after starting the light irradiation? In the time extension experiment in the dark, did it mean no need to irradiate light for the highly efficient removal of MB? Is there any degradation of GO/TiO2/Bent sponge after the light irradiation, such as weight loss of the sample? How about the light irradiation result of the removal of MB by GO/TiO2 sample with bentonite (not sponge)? Figure 3: It is difficult to recognize the absorption spectrum of bentonite. Figure 11: Why were the intensities of PLs of samples compared in an arbitrary unit? Is it appropriate? Manuscript: An appropriate space between words should be used. It is difficult to read, such as “EfficientRemoval” in the title and “Anultrahighly” in the Abstract. References, Journal names: Journal names should be corrected. For example, compare Red. No. 15 and No. 17. References, Ref. No. 13, 14, 16: incomplete page numbers for No. 13 (1175 “-1179“), No. 14 (2741 “-2746“), and No. 16 (7903 “-7907“).
Reviewer 2 Report
This article is an interesting report of the improvement of photocatalytic performance of a graphene-bentonite-TiO2 composite. The aspects that require improvement are minor and are mostly related with the absence of gaps between words all over the article. Some examples of this can be found at:
Lines 2,3,11 Spacing
Line 118 Bentsponge
Line 165 Bentare
Further to this, the following aspect requires the attention of the authors:
Lines 220-223 “Here, to better present the effectiveness of photocatalysis, the concentration of MB is specifically raised to 100 mg/L (Figure7), which is ten times the original MB concentration shown in Figure 6. It can be seen from Figure7 that the adsorbent of 10% GO/TiO2/Bent sponge presents not only a high adsorption effect on MB, but also a highly photocatalytic efficiency under light.”
The authors clearly want to show photocatalytic efficiency with higher MB concentration. You could include the degradation profiles of all other materials.
Reviewer 3 Report
The manuscript shows the removal of MB using GO/TiO2/Bent Sponge and tests the results through various characterization techniques like XRD, SEM, UV, XPS, etc.
The design of experiments is good. However, there are major problems with this paper, like poor writing, both grammatically and technically, the introduction is not extensive and complete, a number of technical flaws, lack of references, etc. Author must address the following issues before this paper is taken into further consideration.
English and the use of propositions should be corrected by taking the help of a native speaker or an expert in English.
Line 60 about sponge... there is no denying that the Authors group has done work on GO sponges.... but just mentioning their own work and implying that this is the only work that has been done (as per the GO sponge introduction) is not a good idea for a scientific paper... The author must mention other publications in support.
L- 146 to 164.. The author has just mentioned the wavenumber assigned for various bong vibrations (textbook knowledge) but has not explained how it adds to this present work specifically..
Fig-2.. FTIR X-axis in the present case is not wavelength.. it is inversely proportional to wavelength.. referred to as wavenumber or frequency...
Line 174, complete the sentence.... "As a result, solar light is utilized more efficiently by 10% GO/TiO2/Bent. " ... more efficient than what?
Line 189- why diffraction pattern of TiO2 so strong? is it because of the high concentration of TiO2 than GO or because it is relatively more crystalline...?
XPS is a surface characterization technique and the results show elemental changes at the sponge outer surface. What about the activity in the large areas of the sponge cavity which comprises the majority of the area? Will the elemental change be the same as they are on the outside of the sponge or not? Did the author broke open these sponge for some of the XPS characterizations or just relying on the outer surface?
Reviewer 4 Report
Before final acceptance the manuscript needs an extensive revision (major revision). Below are my revision comments for the authors: 1. Extensive english correction is required. The authors have not paid enough attention while writing the manuscript. Too many typos were found. 2. Authors should report BET surface area values of each catalysts synthesized herein this work to co-relate with the photocatalytic dye degradation activity. 3. In XRD results, in-spite of using P25, why didn't the authors observe the rutile phase? The data needs to checked again. 4. Stability test: The authors should report how much stable is their photocatalyst by repeated testing of same catalyst? 5. SEM image after photocatalytic tests should also be reported. 6. The authors needs to discuss few more articles pertained with TiO2-Graphene and methylene blue dye degradation. Energy Environ. Sci., 2018, 11, 3183-3193 doi: 10.1039/C8EE00983J; Journal of Photochemistry and PhotobiologyA:Chemistry 358 (2018) 432–440433, doi: 10.1016/j.jphotochem.2017.10.030Author Response
Please see the attachment.

Round 2
Reviewer 3 Report
Figure 5, scale-bars and their measure (e.g. 40um) are not visible. The author should mark them white or at least mention their measurements in the caption if changing the color is difficult for them.
Point 7 XPS, author reply suggests that there is an explanation of XPS broken open samples marked in red in the revised manuscript. I could not find any such text. It is an important point and unless I have somehow missed the explanation, the author is suggested to come back after adding the appropriate explanation.
Reviewer 4 Report
The authors have replied to the reviewer's comments in a satisfactory manner. But there are still few minor typos in the manuscript. Line 37.
Also, journal abbreviation for reference 4, 9 and 10 is incorrect. The authors must correct them
